# SSR-Sequencing Reveals the Inter- and Intraspecific Genetic Variation and Phylogenetic Relationships among an Extensive Collection of Radish (*Raphanus*) Germplasm Resources

**DOI:** 10.3390/biology10121250

**Published:** 2021-11-30

**Authors:** Xiaoman Li, Jinglei Wang, Yang Qiu, Haiping Wang, Peng Wang, Xiaohui Zhang, Caihua Li, Jiangping Song, Wenting Gui, Di Shen, Wenlong Yang, Bin Cai, Liwang Liu, Xixiang Li

**Affiliations:** 1Key Laboratory of Biology and Genetic Improvement of Horticultural Crops, Institute of Vegetables and Flowers, Chinese Academy of Agricultural Sciences, Ministry of Agriculture and Rural Affairs, Beijing 100081, China; LXM911001@163.com (X.L.); wangjinglei@zaas.ac.cn (J.W.); qiuyang@caas.cn (Y.Q.); wanghaiping@caas.cn (H.W.); pengwang0213@gmail.com (P.W.); zhangxiaohui01@caas.cn (X.Z.); songjiangping@caas.cn (J.S.); dshen2016@163.com (D.S.); yangwenlong@caas.cn (W.Y.); 2National Key Laboratory of Crop Genetics and Germplasm Enhancement, Key Laboratory of Horticultural Crop Biology and Genetic Improvement (East China) of Ministry of Agriculture and Rural Affairs, College of Horticulture, Nanjing Agricultural University, Nanjing 210095, China; 3Institute of Vegetables Research, Zhejiang Academy of Agricultural Sciences, Hangzhou 310021, China; 4Genesky Biotechnologies Inc., Shanghai 201315, China; lch@geneskies.com (C.L.); guiwt@geneskies.com (W.G.); caibin@geneskies.com (B.C.)

**Keywords:** *Raphanus* L., genetic diversity, genetic structure, gene flow, evolutionary relationship, SSR-seq

## Abstract

**Simple Summary:**

*Raphanus* is an important genus of Brassicaceae and has undergone a lengthy evolutionary process. However, the inter- and intraspecific phylogenetic relationships and genetic diversity are not well understood. To elucidate these issues, we SSR-sequenced 939 wild, semi-wild and cultivated accessions, and discovered that Europe was the origin center of radishes with diverse European wild radishes, and Europe, South Asia and East Asia might be three independent domestication centers. There was considerable genetic differentiation within European cultivated radishes. European primitive cultivated radish exhibited gene flow with black radish/oil radish and rat-tail radish. Among Asian cultivated radishes, rat-tail radish was a sister to the clade of Chines big radish (including Japanese wild radish), suggesting that they may share the most recent common ancestry. Japanese wild radish had strong gene exchange with Japanese/Korea big radish, oil radish and rat-tail radish. American wild radish developed from natural hybridization between European wild radish and European small radish. All these demonstrated that European primitive cultivated type, American wild radish and Japanese wild radish might have played indispensable roles in radish evolution. Our study provides new perspectives into the origin, evolution and genetic diversity of *Raphanus* and facilitates the conservation and exploitation of radish germplasm resources.

**Abstract:**

*Raphanus* has undergone a lengthy evolutionary process and has rich diversity. However, the inter- and intraspecific phylogenetic relationships and genetic diversity of this genus are not well understood. Through SSR-sequencing and multi-analysis of 939 wild, semi-wild and cultivated accessions, we discovered that the European wild radish (EWR) population is separated from cultivated radishes and has a higher genetic diversity. Frequent intraspecific genetic exchanges occurred in the whole cultivated radish (WCR) population; there was considerable genetic differentiation within the European cultivated radish (ECR) population, which could drive radish diversity formation. Among the ECR subpopulations, European primitive cultivated radishes (EPCRs) with higher genetic diversity are most closely related to the EWR population and exhibit a gene flow with rat-tail radishes (RTRs) and black radishes (BRs)/oil radishes (ORs). Among Asian cultivated radishes (ACRs), Chinese big radishes (CBRs) with a relatively high diversity are furthest from the EWR population, and most Japanese/Korean big radishes (JKBRs) are close to CBR accessions, except for a few old Japanese landraces that are closer to the EPCR. The CBR and JKBR accessions are independent of RTR accessions; however, phylogenetic analysis indicates that the RTR is sister to the clade of CBR (including JWR), which suggests that the RTR may share the most recent common ancestry with CBRs and JWRs. In addition, Japanese wild radishes (JWRs), (namely, *R. sativus* forma *raphanistroides*) are mainly scattered between CBRs and EPCRs in PCoA analysis. Moreover, JWRs have a strong gene exchange with the JKBR, OR and RTR subpopulations. American wild radishes (AWRs) are closely related to European wild and cultivated radishes, and have a gene flow with European small radishes (ESRs), suggesting that the AWR developed from natural hybridization between the EWR and the ESR. Overall, this demonstrates that Europe was the origin center of the radish, and that Europe, South Asia and East Asia appear to have been three independent domestication centers. The EPCR, AWR and JWR, as semi-wild populations, might have played indispensable transitional roles in radish evolution. Our study provides new perspectives into the origin, evolution and genetic diversity of *Raphanus* and facilitates the conservation and exploitation of radish germplasm resources.

## 1. Introduction

*Raphanus*, a member of the Brassicaceae family, consists of two species: *Raphanus sativus* L. and *R. raphanistrum* L. [1]. *R. raphanistrum* L. is a wild relative species of *R. sativus* L., which is an important root vegetable crop species worldwide, comprising five different varieties; namely, *R. sativus* var. *radicular* (European small radish) and *R. sativus* var. *longipinnatus* (East Asian big radish), var. *niger* Kerner (black radish), var. *oleiformis* (oil radish) and var. *caudatus* Hooler & Anderson (rat-tail radish) [2,3,4]. Some previous molecular studies have shown that the wild radish includes three species: *R. raphanistrum*, *R. landra* and *R. maritimus* [5,6]. Other researchers consider that these wild *Raphanus* species should be a single species with three subspecies: *R. raphanistrum* subsp. *raphanistrum*, *R. raphanistrum* subsp. *landra* and *R. raphanistrum* subsp. *maritimus* [7]. However, *R. raphanistrum* L. is generally considered to comprise two subspecies (*R. raphanistrum* subsp. *raphanistrum* and *R. raphanistrum* subsp. *landra*) [8]. Compared with cultivated radish plants, the wild radish has distinct morphological traits, including non-fleshy roots, yellow or white flowers, non-shattering mature siliques containing 1~10 seeds, and strong growth habits influencing crop yields as weeds [9,10,11].

The wild ancestor of the cultivated radish was considered to be *R. raphanistrum* L. by some authors [6,12,13,14]. The cultivated radish has also been proposed to have arisen from hybridization between *R. landra* and *R. maritimus* by other authors [2,15]. Although there are no detailed archeological records of the early history of radish cultivation, the radish may have been domesticated from Europe in pre-Roman times [16]. The wild radish (*R. raphanistrum*) has also been recorded in the Flora of China [17]. The Middle Eastern, Eastern Mediterranean, and Asian regions are believed to be among the multiple centers of radish origin on the basis of chloroplast variation [6]. A recent molecular study suggested that the wild radish originated from the Mediterranean region, whereas the cultivated radish originated in the West, migrated to South and Southeast Asia, and then spread independently from South and Southeast Asia to China and Japan, according to the DNA sequence variation over 500 radish accessions from East Asia, South and Southeast Asia, in addition to the Occident and Near East, by double-digest restriction site-associated DNA sequencing (ddRAD-Seq) technology [14]. Asian cultivars were typically different from Mediterranean (European) cultivars and were closely related to wild Asian accessions rather than wild radish accessions from Europe [6]. In Japan, a form of wild radish germplasm was found being typically distributed along the coast; Makino inferred that this Japanese wild radish “escaped” from the cultivated radish and classified this wild radish as *Raphanus. sativus* forma *raphanistroides* [18]. This hypothesis is generally accepted but has not been confirmed through studies on mtDNA variation [19,20]. *R. sativus* var. *raphanistroides* is also widely distributed along the coast of China and Korea [18]. Kim revealed that the wild relative of Asian cultivars might be *R. sativus* var. *raphanistroides*, rather than the wild *R. raphanistrum* [21]. The California wild radish was thought to originate from hybridization between cultivated radishes and wild radishes (*R. raphanistrum*) less than 150 years ago [22,23]. To date, even though the main aspects of the origin and evolution of radishes worldwide have been uncovered, much remains unclear. Specifically, the origin of Asian cultivated radishes, their genetic structure, genetic diversity, and relationships with other taxa.

Genetic diversity is the basis of radish evolution and differentiation. Wang et al. [4] used 50 core-expressed sequence-tag (EST)-SSR markers to evaluate 83 cultivated radish genotypes, 3 accessions of East Asian wild radish (*R. sativus* var. *raphanistroides* Makino), 7 accessions of *R. raphanistrum*, with an average diversity of 4.88 alleles per locus and a polymorphism information content (PIC) value of 0.55. Other researchers used 221 amplified fragment-length polymorphism (AFLP) markers to assess the diversity of 65 cultivated radish accessions from 21 countries in Europe, North Africa, the Middle East, and South and East Asia, and found Nei genetic diversity values ranging from 0.267 to 0.279 [24]. Kong also used 8 pairs of AFLP primers to analyze 56 landraces and commercial cultivars from different regions and countries, 327 bands were detected and 39.1% were polymorphic [25]. Japanese researchers conducted a ddRAD-seq and detected 52,559 single-nucleotide polymorphisms (SNPs) over 500 radish accessions, including 502 accessions of *R. sativus*, 13 accessions of *R. raphanistrum*, 3 accesions of *R. maritimus*, and 2 accessions unknown from East Asia, South and Southeast Asia, and the Occident and Near East [14]. Unfortunately, none of the abovementioned studies focused on gene diversity or the diversity distribution among an extensive collection of different radish species, subspecies and varieties.

SSRs are 2–5 bp sequences that are repeated nose-to-tail; the number of copies at each locus is what varies [26]. Due to their advantageous characteristics of universal distribution, high density and codominance, SSRs have been used as effective tools in genetic diversity analysis [27,28,29,30,31,32,33], gene mapping [34], and genome-wide association studies (GWASs) of major quantitative-trait loci (QTLs) [35]. Traditional gel electrophoresis is unable to accurately distinguish the number of repeats present in SSR amplicons. Compared with previous SSR marker techniques, SSR-sequencing (SSR-seq) is a new high-throughput, accurate, rapid, and reasonably priced technique that produces highly polymorphic results [36]. and has been successfully applied to grapevine (*Vitis vinifera* L.) [37], cucumber (*Cucumis sativus* L.) [38], *Salmo salar*, *Quercus faginea*, *Quercus canariensis*, *Alosa alosa*, *Armillaria ostoyae*, *Melipona variegatipes* [39], and hexaploid *Camellia oleifera* [40].

In this study, a total of 939 radish accessions were subjected to SSR-seq with 38 core SSR markers, which were designed based on the radish whole genome. Based on the diversity of the SSR markers, the genetic diversity and phylogenetic relationships among the radishes were further analyzed. The results provide valuable information on the genetic diversity distribution, origin and evolutionary trajectory of the radish and lay a solid foundation for effective germplasm collection and preservation, elite gene mining, and genetic improvement in the radish.

## 2. Materials and Methods

### 2.1. Material Planting

A total of 939 cultivated and wild radish accessions were used in this study. These accessions were conserved in the National Mid-term Genebank of Vegetable Germplasm in China and collected from 31 provinces of China and from other countries, such as Japan, Korea, Thailand, India, Germany, Russia, Spain, France and the United States (Appendix A). The cultivated germplasm resources included the black radish (BR), the European small radish (ESR), the rat-tail radish (RTR), the oil radish (OR), the East Asia cultivated radish (EACR) (including the Chinese big radish (CBR) and the Japanese/Korean big radish (JKBR)) and the European primitive cultivated radish (EPCR) accessions. Here, the EPCR was composed of two parts: eighteen accessions were recorded as *R.*
*raphanistrum* but had characteristics of taproots, flowers, and siliques that were similar to those of the cultivated radish; the other forty-two accessions with the Latin name of *R. sativus* L. convar. *sativus* had characteristics of non-enlarged taproots, white or purple flowers, and hard siliques that were between those of cultivated radishes and European wild radishes (EWR). Among the accessions named wild radishes, the Japanese wild radish (JWR), the American wild radish (AWR; also known as California weedy radish) and the EWR, were included. Two subspecies of EWR, *R*. *raphanistrum* L. subsp. *raphanistrum* (72 accessions) and *R*. *raphanistrum* L. subsp. *landra* (Moretti ex DC.) Bonnier & Layens (118 accessions), as well as 11 accessions of wild radish with incomplete Latin names according to their original records, were included. All the seeds were planted in plastic greenhouses on Langfang Farm in the spring of 2017 and autumn of 2018 and cultivated normally, according to commercial production standards. We planted ten plants for each accession. 

### 2.2. Phenotypic Observations

To verify the botanical classification of all the accessions, their major phenotypic traits were observed and recorded during the vegetative growth and reproductive stages, according to the descriptors and data standards for the radish (*Raphanus sativus* L.) [41]. The vegetative organ traits included: leaf shape; shape of leaf apex; leaf pubescence; leaf incision; fleshy root shape; distribution of the lateral root, and the exterior color of fleshy taproot. The reproductive traits included: flower color; flowing time; silique surface; silique setting posture, and silique shattering habit. 

### 2.3. DNA Extraction

We collected the fresh and tender leaves from three normal and representative 2-week-old seedlings of each accession and pooled them together as a sample. Total genomic DNA was extracted for each accession using the modified cetyl-trimethylammonium bromide (CTAB) method [42]. We added about 200 mg of the fresh leaves, two steel balls (4 mm in diameter), and 700 uL CTAB into the centrifuge tube together, then used the crusher to break the sample into powder. Then, we put the tube into 65 °C water for water bath heating. The DNA quality was assessed using a Nanodrop 2000 spectrophotometer: the purity was strictly controlled for OD-specific values of 260/280 between 1.7 and 2.0 and 260/230 values of more than 1.8. Additionally, the final concentration of the primers was 0.1~0.3 μM and the concentration of the genomic DNA was not less than 3 ng/μL. Finally, 1% agarose gel was used to assess the quality of the DNA.

### 2.4. SSR Primer Design and Genotyping

Based on the XYB36–2 reference genome [43], MISA and Primer 3 software were used to design more than 600 SSR primers according to their position on each chromosome for an interval of 1 Mb. Twelve radish samples were selected to evaluate the polymorphism of the SSR markers, including two subspecies of EWR, the EPCR, one European and one Chinese oil radish, and the BR, ESR, AWR, RTR, JWR, JKBR and CBR.

The SSR primers should satisfy the requirements of SSR-seq: the unit is repeated from 3 to 10 times, and the repeat unit cannot be composed of only GC or AT. For the screening of polymorphic primers, the PCR amplification reaction mixture consisted of the following: 0.3 μL of each primer; 2 μL of genomic DNA; 1.5 µL of 10X buffer; 0.3 µL of dNTPs (2.5 mmol/L); 0.2 µL of Taq polymerase (2.5 U/µL), and 10.4 µL of deionized water (final reaction mixture volume of 15 μL). The PCR amplification conditions were as follows: 1 cycle of 95 °C for 5 min; 30 to 35 cycles of 95 °C for 35 s, 55 °C for 30 s, and 72 °C for 1 min; and 1 cycle of 72 °C for a final extension of 7 min. The amplification products were separated by 8% polypropylene gel electrophoresis for 1 h at 200 V to assess the sample polymorphism. Polymorphic primers were used to amplify the DNAs of all materials and the amplified products were detected and genotyped on an Illumina NextSeq instrument (Illumina, San Diego, CA, USA).

### 2.5. Analysis of Genetic Diversity and Phylogenetic Relationships among Different Populations

The genetic diversity among different populations and across all the populations was estimated based on SSR markers according to the Na (observed number of alleles), Ne, (effective number of alleles), Ho (observed heterozygosity), He (expected heterozygosity), I (Shannon’s information index) and Nei indexes. All these indexes were determined using normalized SSR genotyping data with POPGENE v1.32 software.

Based on the microsatellite data and the Bayesian model, STRUCTURE v2.3.4 software was used to determine the optimum number of subpopulations for all the analyzed samples. For each *K* value ranging from 1 to 11, the STRUCTURE program was run 11 times with an admixture model and a Markov chain Monte Carlo (MCMC) burn-in length of 100,000, after which 50,000 burn-in runs and MCMC replicates were set up. Finally, the largest ΔK corresponding to the optimal *K* values was obtained based on the results of the ΔK analysis.

The Nei genetic distance was calculated based on the allele frequencies of subpopulations, and the genetic similarity matrixes and genetic distance matrixes were obtained between subpopulations. Finally, the FastME method in the phangorn R package was used for the construction and rendering of an evolutionary tree. The polysat package was run in R 3.5 software and used to calculate the Bruvo distance between each pair of samples and to perform PCoA [44]. Weir and Cockerham’s F-statistics (Fis, Fit, Fst) and Nm for all the samples and each population of wild and cultivated radish accessions were also estimated via POPGENE software. TreeMix v1.13 software [45] was subsequently used to estimate the gene flow among the different radish germplasm resource categories.

## 3. Results

### 3.1. Classical Classification of the Radish Germplasm Collection Based on Basic Phenotypic Characteristics

According to the observation of phenotypic characteristics and original records, the radish collection showed rich morphological diversity in shape, the size and color of the fleshy taproot, leaves, flowers and siliques (Appendix A). We divided all the genotypes into ten categories: EWR (201 accessions); EPCR (60); OR (16); BR (5); ESR (18); AWR (3); RTR (26); JWR (17); JKBR (47) and CBR (546) (Appendix A). The EWR population included two subspecies with well-developed lateral roots and unexpanded taproots. Among them, subsp. *raphanistrum* accessions bloomed and produced seeds in the fall of the same planting year in Beijing. There were many thorns on the flower buds, leaves and stems. The silique surface was constricted between seeds. As the silique matured, it could automatically break at the constriction, and the seeds with hard shells were easily shattered. The taproot of these accessions was not enlarged and had highly branched lateral roots. With respect to subsp. *landra*, most of the accessions rarely bloomed in the autumn of the planting year in Beijing, but produced yellow flowers in the following spring. These accessions had a few thorns on their flower buds, leaves and stems. The silique surface of most accessions shrank between seeds; however, for some accessions it was smooth or wavy. Mature siliques either shattered later or did not shatter. Compared with those of subsp. *raphanistrum*, the unenlarged taproots of subsp. *landra* had fewer lateral roots.

In addition, eighteen accessions named *R. raphanistrum* from Europe had semi-enlarged and severely lignified taproots with an abundance of lateral roots, white or purple flowers and non-shattering siliques. We considered these to be members of the EPCR population because they resembled the weedy radish. At the same time, forty-two accessions with the Latin name of *R. sativus* L. convar. *sativus,* as primitive cultivated radishes, had the characteristics of non-enlarged or semi-enlarged and lignified taproots, white or purple flowers, and hard, lignified siliques. We also classified them into the EPCR population.

Most of the OR accessions had semi-enlarged lignified taproots with many lateral roots and many leaves. However, the Chinese oil radish had non-enlarged taproots with many lateral roots. The BR accessions with black, long, cylindrical or round fleshy taproots were collected from Europe. ESR primarily had red, purple and white skin and round or flat-round taproots. The RTR accessions had non-enlarged or semi-enlarged taproots and easily bolted. Their tender siliques with seeds could be eaten as the main edible products, and the largest silique reached 32.89 cm.

The AWR accessions had non-enlarged taproots, as well as red, purple or white flowers, with a few thorns on their stems, leaves and flower buds. The characteristics of the JWR accessions included: white, light red and purple flowers; few thorns on their stems, leaves and flower buds; non-enlarged or semi-enlarged taproots with many lateral roots; and highly hard, lignified mature siliques with wavy surfaces. Their seeds were not easy to burst or thresh. Most of the JKBR accessions had long, cylindrical, fleshy taproots with white, light green or red skin. However, Kosena daikon, Moriguchi daikon and Yamada daikon in the JKBR population had enlarged taproots with some lateral roots. CBR accounted for most of the materials and had enlarged taproots with a variety of colors, sizes and shapes. All of the above information provides the basis for further analysis of this collection at the molecular level.

### 3.2. Screening of Core Primers for SSR-Seq

In total, 600 pairs of SSR primers distributed throughout the radish whole genome were preliminarily designed. Genotyping was ultimately conducted by amplifying the template of DNA using 38 pairs of highly stable polymorphic SSR primers (Appendix A), which were relatively evenly distributed across the nine radish chromosomes and qualified for SSR-seq (Appendix A). Different primers have different resolutions and roles in distinguishing the genetic variation in radish germplasm resources (Appendix A). The number of alleles (Na) of all accessions ranged from 3 for primer RS2–37, to 22 for RS8–15. The effective number of alleles (Ne) ranged from 1.90 (RS4–33) to 5.97 (RS9–12). The lowest observed heterozygosity (Ho) was observed for RS2–37 (0.10), whereas the greatest was 0.74 (for RS1–21). The expected heterozygosity (He) varied from 0.47 (RS4–33) to 0.83 (RS9–12), Shannon′s information index (I) ranged from 0.82 (RS4–1) to 2.11 (RS9–12), and the Nei index varied from 0.47 (RS4–33) to 0.83 (RS9–12).

### 3.3. Genetic Diversity of Radish Germplasm Resources

In total, 424 alleles were detected in all the accessions by SSR-seq. We analyzed the genetic diversity of ten groups based on the allele variation profile (Table 1). The EWR accessions had the highest Na (9.34), Ne (4.40), He (0.71), Nei (0.70) and I (1.59) indexes, indicating that Europe may have been the earliest center of origin of the radish. The AWR accessions showed the lowest Na (1.92), Ne (1.75), He (0.42), Nei (0.34) and I (0.52) indexes, indicating that their genetic variation was relatively narrow. Compared with the EWR accessions, the JWR and BR populations had moderate genetic diversity. The JKBR, RTR, ESR and OR populations exhibited a moderate to high genetic diversity, according to their Na, He, Nei and I index. Notably, most indexes for the EPCR and CBR populations were on the high side. The Na and Ho index for CBR accessions were higher than those for EPCR accessions, whereas the Na, He, Nei and I index for the EPCR accessions were higher than those for the CBR accessions. We also calculated the genetic diversity indexes of the Asian cultivated radish (ACR) and European cultivated radish (ECR) populations, which showed that the Na and Ho index of ACR were higher than those of the ECR accessions, but that the Ne, He, Nei and I index of the former were lower than those of the latter. These findings imply that Europe, South Asia and East Asia may have been different domestication centers of the radish.

### 3.4. Genetic Structure of Radish Germplasm Resources

To investigate the population structure of the collected radish genotypes, the Bayesian clustering program STRUCTURE based on SSR fragment length was used. ΔK analysis showed K = 2 to be optimal for all radish samples (Figure 1A), indicating that all the accessions can be distinctly classified into two groups: cultivated radish and wild radish. We arranged all the materials in the different categories according to their botanical classification and compared their genetic composition (Figure 1B). A considerable portion of germplasm resources in the AWR, ESR, OR, BR, EPCR, RTR, JKBR and JWR categories contained admixtures of cultivated radish and wild radish genetic backgrounds, and the ESR, AWR, OR, BR and EPCR accessions were more highly composed of wild radish, indicating that these genotypes may have been more primitive than the other genotypes.

### 3.5. Phylogenetic Relationships among Radish Germplasm Resources

A neighbor-joining phylogenetic tree was constructed to show the phylogenetic relationships of all cultivated and wild radish genotypes (Figure 2). The EWR and CBR accessions were the most distantly related (located at the two ends of the phylogenetic tree). EWRs were divided into two subgroups. The first subgroup included 66 radish accessions, among which 63 were recorded as subsp. *raphanistrum*; two accessions were named as subsp. *landra*, and one accession had an incomplete Latin name according to the original records. Similarly, the second subgroup included 135 radish accessions, among which 116 accessions are recorded as subsp. *landra*; 9 accessions were called subsp. *raphanistrum*, and the Latin names of 10 accessions were incomplete. Combined with the findings based on the phenotypic observations, 3 accessions in the first subgroup and 19 accessions in the second subgroup were corrected to subsp. *raphanistrum* and subsp. *landra*, respectively, which helped to verify the taxonomic status of these controversial radish genotypes.

Notably, among the sixty accessions of EPCR from Europe, forty-two accessions of *R. sativus* L. convar. *sativus* had non-enlarged or semi-enlarged taproots with an abundance of lateral roots, white or purple flowers, hard, lignified siliques with a wavy surface, and non-shattering seeds. Most of these accessions were relatively closely related to the EWR accessions and seemed to be more primitive than the other cultivated radish accessions. A few of them were distributed among the RTR and EACR populations, implying that they may have been the transitional ancestors of RTR and EACR, respectively. The Latin names of the other eighteen radish accessions were recorded as *R. raphanistrum* subsp. *raphanistrum* and *R. raphanistrum* subsp. *landra.* However, their phenotypic traits were similar to those of the forty-two materials mentioned above, and they clustered closer to the EWR population. Therefore, the genotyping supported our classification of these accessions into the EPCR population.

Three accessions of the AWR were scattered among those of the ECR, although they had more genetic components resembling the wild radish accessions. Among them were 247Y and 249Y, which had dark purple flowers and were clustered together, and another (248Y) had white flowers separated. Thirteen ESR accessions had white or light red flowers and most of them were clustered together, whereas the accession 602C was a unique ESR germplasm and had light yellow flowers, which were clustered into a separate branch. Five BR accessions had the same origin and were clustered together. Moreover, most of the OR accessions from Europe were clustered together, except for two accessions collected from Yunnan Province, China, which clustered far from the European OR. Almost all the RTR accessions introduced from Germany and India clustered together. The CBR accessions, accounting for 58.15% of all accessions, were collected from 31 provinces of China and clustered together into a large subgroup. Most of the JKBR accessions were closely related to the CBR population, and a few old Japanese landraces were close to the EPCR.

The seventeen JWR accessions were spread among the EACR and South Asia cultivated radish (SACR) accessions, which showed that they were phenotypically diverse. Our results essentially agreed with the viewpoint of Yamagishi [46] and Kim [21], that JWR accessions belong to cultivated radishes from the phylogenetic clustering tree; however, we suggest that they have a different variation from the EACR and SACR populations, because they carry more wild radish genetic background, according to the genetic structure analysis. Furthermore, this clustering result indicated that the RTR is a sister to the clade of the CBR (including JWR), which suggested that the RTR may share the most recent common ancestry with the CBR and JWR.

The PCoA of the SSR data revealed that in terms of phylogenetic relationships, the EACR and EWR were also clearly separated far away in the horizontal direction (Figure 3). Most of the ECRs (BR, ESR, OR) were parted with the EWR in the vertical direction. The EPCR and RTR accessions were generally distributed between the EWR and the EACR in somewhat different directions. However, most of the RTR accessions were close to the EACR, whereas the BR, OR and ESR were close to each other, and relatively close to the EPCR and EWR accessions.

Most of the JWR accessions were distributed among or near the EACR accessions, although some were between the EACR and EWR accessions (Figure 3). These results are slightly different from the results in Figure 2, suggesting that the JWR accessions may have been the intermediate type in evolution or domestication. Combined with their phenotypes, we consider the JWR to be a type of semi-wild resource in further differentiation. The AWR accessions were dispersed among the European cultivated radish and the EWR accessions, confirming a certain relationship between them.

### 3.6. Population Differentiation and Gene Flow of Radish Germplasm Resources

Population differentiation within and among the different radish populations was evaluated through Wright’s (1965) F-statistics, the inbreeding index (Fis), and the Fit and fixation index (Fst) (Table 2). The overall population (OP) was assigned to ten different categories according to the botanical classification, as described above. The Fis values of the OP (0.60) whole cultivated radish (WCR: includes ESR, OR, RTR, BR, EPCR, JKBR and CBR) populations (0.58) and the ECR (0.56) populations were approximate, demonstrating similar inbreeding frequency among and within species and cultivated materials. The ACR population had the lowest Fis value (0.46), which may indicate that the EACR and SACR populations had less gene exchange in Asia, and that these two populations were domesticated independently. The Fst value (0.28) of the ECR population was the highest, followed by the Fst value of the OP (0.21), indicating that a considerable genetic differentiation occurred in the ECR accessions, which played an indispensable role in OP differentiation. The number of migrants per generation (Nm) was the highest for the WCR population (1.17), indicating a high degree of intraspecific gene exchange.

Gene flow among different species/populations that typically occurs in ecosystems is helpful for understanding genetic exchange and species diversification. In terms of gene flow strength among different radish species, varieties or types (Figure 4), we found that the JKBR accessions have a strong gene flow with the JWR accessions, indicating that accessions of the JWR, a semi-wild type of cultivated radish, might frequently crosspollinate with cultivated radish accessions in the same habitat, along coastal areas of Japan. In addition, the JWR accessions exhibited evidence of gene flow with members of the OR and RTR populations, implying that genetic exchange occurred between the JWR and Chinese/European oil radishes, and between the JWR and European/Indian rat-tail radishes during certain periods and in specific situations, under which they once coexisted. Moreover, gene flow occurred not only between the EPCR and the South Asia rat-tail radish, but also between the EPCR and BR accessions at the early stages of cultivated radish evolution, possibly due to their similar range in Europe or their historical colonial relationship, indicating that the EPCR holds an important position in radish evolution. Combining the gene flow and phylogenetic relationship data, we inferred an indirect relationship between the EPCR and JWR during the long process of natural variation/crossing and artificial domestication.

Interestingly, no gene flow was detected between the OR population and the RTR population, although members of both populations are eaten as siliques or used as oil sources. This lack of gene flow may have occurred because of geographical or cultural isolation. In addition, the AWR accessions showed evidence of gene flow with the ESR accessions, suggesting that the AWR may be the offspring of natural hybridization between the ESR and *R. raphanistrum*.

By combining the data from the phylogenetic clustering and PCoA, we inferred that EPCR, AWR and JWR populations might have played a special transitional role during the course of radish evolution. In Europe, a possible evolutionary path might be from the EWR population to the EPCR, and then to cultivated radishes of the BR, ESR and OR populations. In South Asia, the path might have been from the EWR or EPCR, to the JWR or RTR populations. In East Asia, the RTR is the sister to the CBR/JWR clade, suggesting that they share the most recent common ancestor. This indicates the direction of evolution from the EPCR to an ancestor of the RTR and JWR/CBR, before the divergence of these two groups separated geographically: then, to the Chinese oil radish/Japanese traditional landraces and finally to the Asian cultivated big radish.

## 4. Discussion

The origin, evolution, and classification of the radish has been debated by many researchers [6,14]. Nonetheless, little attention has been given to the status of genetic diversity and genetic exchange among different species, varieties and accessions of radishes [47]. In this study, an effective SSR-seq technique based on 38 pairs of polymorphic SSR core primers was the first to be successfully applied to genotype 939 radish accessions, which is the largest scale to date. Based on these findings, we discovered new evidence supporting the taxonomy and evolutionary relationship of the radish.

### 4.1. Development and Application of SSR-Seq Technology Suitable for Genotyping Radish Germplasm Resources

Traditional SSR markers, which are highly polymorphic and codominant, have been widely used in genetic diversity evaluations [27,29,30,31,32,33], QTL mapping [35], background selection and marker-assisted selection (MAS)-based breeding [48]. The advantages of SSR-seq technology in relation to the use of conventional SSRs and other markers have been demonstrated [36,40]. In our study, SSR-seq with 38 SSR primers from the radish whole genome was applied to effectively distinguish 939 radish accessions for the first time. This combination of SSR-seq and core primers was highly valuable for exploring the background information on germplasm resources, conducting taxonomic and genetic improvement studies and protecting the breeders’ intellectual property.

### 4.2. Radish Origin and Domestication Centers

During the process of plant evolution and crop domestication, a few agronomically desirable alleles were selected, and numerous genetic variations in wild progenitor species were lost due to natural/artificial selection and bottleneck effects [49], as has been shown in barley [50], soybean [51], watermelon [52], etc. Usually, the areas with the greatest genetic diversity are the centers of origin for, or evolution of, crop plants [53,54,55]. It has been suggested that *R. raphanistrum* originated in the Mediterranean region [56]. Our study revealed that among all the germplasm resources, the EWR was the most genetically diverse. In addition, we found that the ECR and EACR had approximately similar genetic diversity. Among the cultivated radish categories, the EPCR and CBR accessions were more genetically diverse than the accessions in the other cultivated categories. Moreover, the RTR, the genetic diversity of which is moderate, experienced gene flow with the JWR and EPCR populations. Taken together, these findings confirm that Europe is the radish center of origin and that Europe, South Asia and East Asia may have been independent domestication centers of radish.

### 4.3. Phylogenetic Relationships among Radish Germplasm Resources and Possible Evolutionary Paths

Previous findings on the evolution of cultivated radish accessions are controversial [1,2,21,57], which may be due to the limitations in the number and background information of germplasm resources and molecular markers [3]. The subsp. *raphanistrum* and subsp. *landra* are recorded in our collection. Our phenotypic observations matched the characteristics [17] of the two subspecies. Accessions of subsp. *raphanistrum* have yellow flowers and many thorns on their flower buds, leaves and stems. In addition, the plants flower easily, the surface of the siliques is shrunken between the seeds, and their lignified and hard siliques easily break between seeds, which then fall automatically. Self-shattering is an important diagnostic trait of domesticated crops and is beneficial for their rapid reproduction, high seed viability and ability to survive in different stressful environments. However, this phenomenon seems to have been lost during cultivation [58]. According to the phylogenetic relations, subsp. *raphanistrum* is located farthest from the cultivated radish. Based on all the evidence, we infer that subsp. *raphanistrum* may be the most primitive wild ancestor of cultivated *R. sativus*.

Based on our observations, members of subsp. *landra* have yellow flowers and few thorns on their flower buds, leaves, stems and silique surface. Additionally, they rarely flower in the autumn of the sowing season. In most cases, the surface of the lignified and hard siliques is shrunken between the seeds, and some are smooth or wavy. When they mature, the siliques usually do not fall off or shatter. Similar phenotypic traits were also observed in the EPCR. However, the flowers of the EPCR plants are white, red and purple, and they bolt late. Previous anatomical data has confirmed that the multilayered palisade tissue of subsp. *landra* is present only in cotyledonary leaves of cultivated radish plants [59]. Our phylogenetic data indicated that subsp. *landra* are closely related to the primitive cultivated radish accessions. Overall, we infer that subsp. *landra* may be the closest wild ancestor to *R. sativus.*

Among the cultivated radish accessions, the degree of genetic differentiation is higher in the ECR accessions, whereas the evidence of gene flow is stronger in ACR accessions. The European and Asian cultivated radish accessions are relatively independent. The BR, ESR and OR accessions are closely related to the EWR accessions, whereas the RTR accessions are closely related to the East Asian radish accessions. EPCR accessions seem to play a bridging role connecting the ECR, SACR and EACR populations. Combining our evidence with previous findings [6,14,60], we speculate that the most likely evolutionary path was as follows: radiating from *R. raphanistrum* in the Mediterranean, the EPCR accessions evolved/domesticated first, after which the OR, ESR and BR accessions appeared successively across Europe. On the other hand, the EPCR might have been transferred to Asia and experienced independent domestication in South Asia (India) and East Asia (China, Japan and Korea). As a result, RTR, CBR and JKBR genotypes were domesticated following the domestication of the Japanese (Chinese) wild radish or oil radish accessions.

### 4.4. Taxonomy of R. Sativus var. Raphanistroides, a Semi-Cultivated Type

JWR accessions are distributed mainly along the sandy coast and are found at numerous latitudes and longitudes in East Asia [61,62]. However, the taxonomic status of the JWR is controversial. Yamagishi and Terachi [19] proposed that the major sequence of *orf138* is distributed within *R. raphanistrum* and JWR accessions; thus, the Ogura male-sterile cytoplasm originated from a wild species and was introduced into JWR [19]. However, Makino assumed that JWR accessions “escaped” from the cultivated radish and were classified into wild radishes as *R. sativus* forma *raphanistroides*, although this hypothesis has not been confirmed [19]. In our study, the JWR accessions clustered mainly together with East Asian and South Asian cultivated radish accessions, according to the dendrogram, but were distributed between the EWR and EACR accessions in the PCoA diagrams, based on the genotyping data. On the basis of morphology, the JWR accessions exhibited semi-enlarged or unenlarged taproots, light red or purple flowers, a wavy surface of siliques and lignified, hard siliques with non-shattering seeds. Based on these characteristics, we infer that the JWR accessions may have constituted an intermediate diverse group during domestication and tended to be from cultivated radish types.

From the perspective of gene flow, JKBR accessions have undergone the strongest degree of gene flow with JWR accessions, indicating that JWR accessions have undergone extensive genetic exchange with cultivated radish accessions in Asian regions, which may have resulted in the close relationship between these populations. Other research has also revealed that JWR plants can fully cross with other cultivars [19]. The JWR populations also exhibit gene flow with OR accessions collected from the Yunnan Province in China and from Germany, explaining why different OR accessions may have similar relationships. These results implied that the JWR, an intermediate germplasm resource, may have played an important role in East Asian radish evolution and the formation of genetic diversity.

### 4.5. The Evolutionary Role of American Wild Radish as a Weed in Competition with Cultivated Crops

Bidirectional gene flow between wild and cultivated species has become a frequent phenomenon, and plant evolution throughout domestication has contributed to plant diversification [58]. If gene flow between two isolated radish taxa occurs through hybridization and biparental introgression, their offspring may attain the same or higher competitiveness, and ultimately replace the parents in nature [23,63]. It has been reported that the AWR (California wild radish) is the offspring of *Raphanus sativus* L. and *Raphanus raphanistrum* L. [22,23]. Three accessions of AWR were clustered in intermediate parts among EWRs and ECRs according to clustering (Figure 2) and PCoA (Figure 3) and exhibited evidence of gene flow with ESR accessions. Our findings are basically consistent with a previous study, showing that the AWR accession was an offspring by back-crossing of the inter-species hybrid of *R. raphanistrum* ssp. *raphanistrum* and *R. sativus* var. *radicula* for two generations, with *R. sativus* var. *radicula* as the recurrent parent through comparative genome analysis [60]. Thus, the AWR may be another source of new radish varieties and types.

In conclusion, SSR-seq of 939 radish accessions revealed that EWR accessions are the most diverse, indicating that Europe is the radish center of origin. The EPCR, CBR and JKBR genotypes are relatively highly genetically diverse, followed by the OR, RTR and ESR accessions, suggesting that Europe, South Asia and East Asia may have been independent domestication centers of radishes. The combination of phylogenetic analysis, genetic structural analysis and PCoA effectively distinguished the EWR and cultivated the radish populations. Each of the cultivated varieties cluster in a relatively concentrated group, with different distributions. The CBR is far from the EWR and close to the JKBR. The RTR is located between the ECR and EACRs. JWR accessions, as semi-wild radishes, are located between those of the EWR and CBR, but most of them are closer to EACR accessions. JWR accessions exhibit gene flow with the RTR, OR and JKBR accessions. EPCR accessions with high diversity are dispersed extensively throughout both EWR and ECR accessions, and some accessions are closely related to Asian accessions. There is evidence of gene flow between the EPCR and RTR accessions, and between the EPCR and BR/OR accessions. Gene flow analysis also indicated that the AWR might be the progeny of the ESR and *R. raphanistrum*. From these discoveries, we inferred that EPCR, AWR and JWR accessions might have played a special transitional role in the course of radish evolution. In Europe, a possible evolutionary path might have been from the EWR to the EPCR, to the BR, ESR and ORs. In South Asia, the path might have been from the EWR to the EPCR, or the JWR to the RTR. In East Asia, the evolutionary path might have been from the EPCR to an ancestor of the RTR and JWR/CBR, to the Chinese oil radish/Japanese traditional landraces to the Asian cultivated big radish. Our study provides a comprehensive understanding of radish diversity, distribution, origin and domestication. The results will facilitate in-depth research and utilization of radish germplasm resources.

## Figures and Tables

**Figure 1 biology-10-01250-f001:**
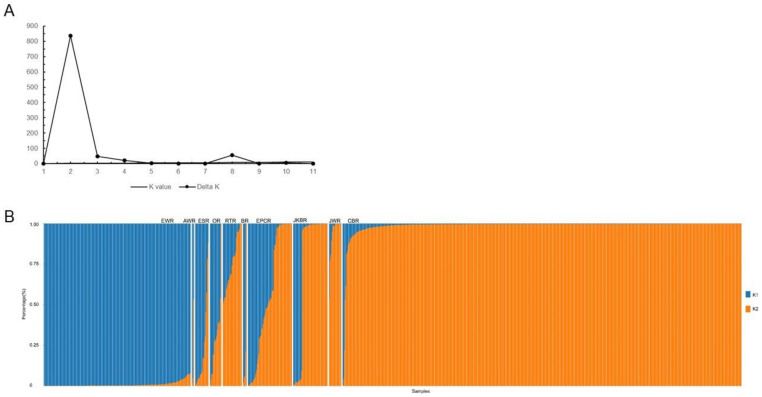
Population structure of 939 radish accessions. (**A**) Optimal *K* value for population structure analysis (K = 2). (**B**) Population structure (K = 2) of all materials. The different colors represent the probability of each accession having a different genetic background. Orange represents the cultivated radish, and blue represents the wild radish.

**Figure 2 biology-10-01250-f002:**
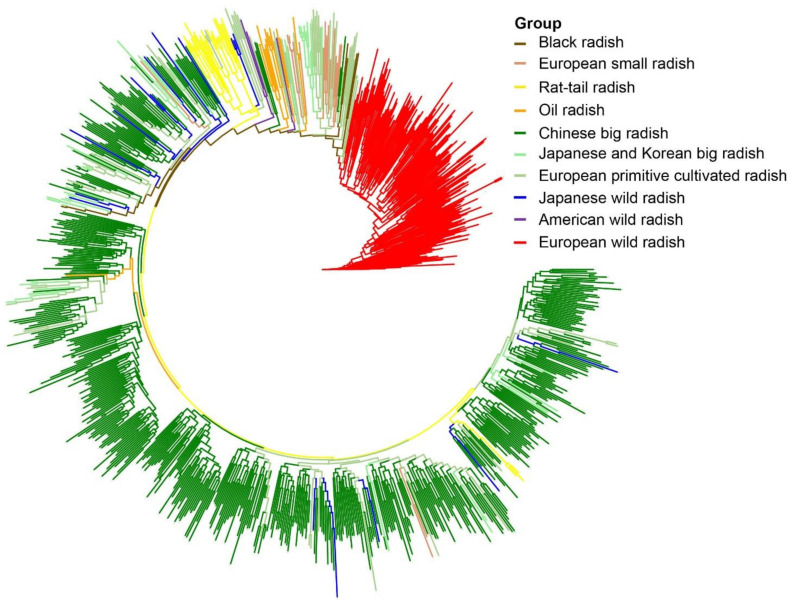
Phylogram of 939 radish genotypes based on SSR-Seq with 38 pairs of SSR primers.

**Figure 3 biology-10-01250-f003:**
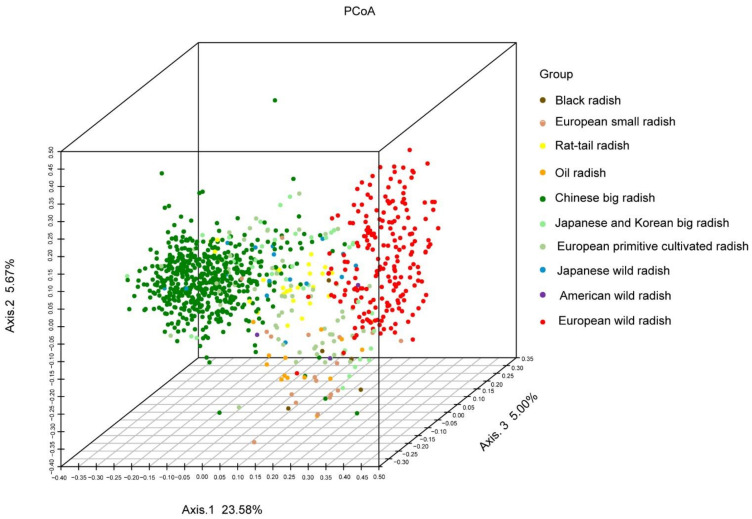
PCoA of 939 radish genotypes based on SSR-Seq with 38 pairs of SSR primers.

**Figure 4 biology-10-01250-f004:**
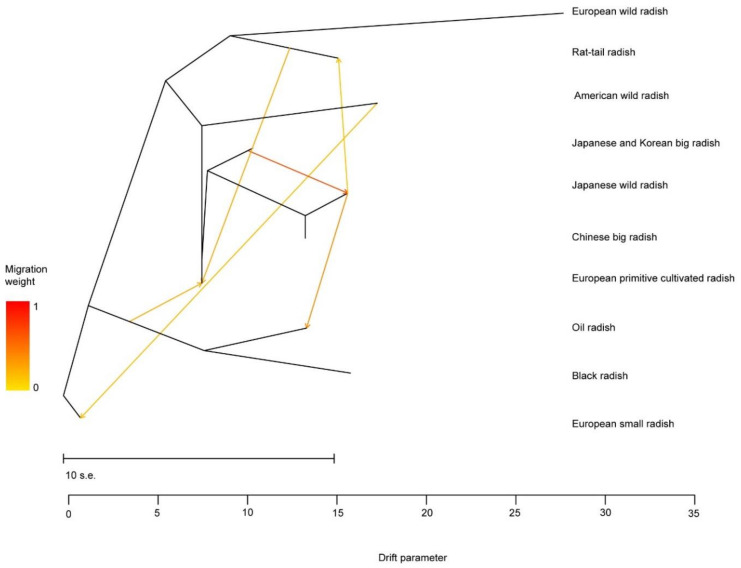
Gene flow among different cultivated and wild radish populations. The color of the bar shows the migration weight, and the arrows indicate the direction of gene flow.

**Table 1 biology-10-01250-t001:** Genetic diversity estimates within different wild and cultivated radish populations.

Index	EWR	AWR	ESR	OR	RTR	BR	EPCR	JKBR	JWR	CBR	ACR	ECR	AR	Max	Min	Average
Accessions	201	3	18	16	26	5	60	47	17	546	618	100	939			
Na	9.34	1.92	4.05	4.32	4.08	2.58	6.63	5.32	3.84	8.08	8.66	7.53	11.16	11.16	1.92	5.95
Ne	4.40	1.75	2.67	2.81	2.39	2.22	3.58	2.81	2.58	2.47	2.56	3.63	3.33	4.4	1.75	2.86
Ho	0.28	0.14	0.17	0.43	0.19	0.12	0.24	0.17	0.22	0.33	0.32	0.24	0.30	0.43	0.12	0.24
He	0.71	0.42	0.59	0.61	0.53	0.53	0.68	0.61	0.57	0.55	0.57	0.69	0.67	0.71	0.42	0.59
Nei	0.70	0.34	0.57	0.58	0.51	0.46	0.67	0.6	0.54	0.55	0.57	0.69	0.67	0.7	0.34	0.57
I	1.59	0.52	1.05	1.10	0.96	0.77	1.39	1.16	1.00	1.08	1.13	1.43	1.45	1.59	0.52	1.12

Notes: Asian cultivated radish: ACR, which includes CBR, JKBR, Chinese oil radish [2] and South Asia rat-tail radish [23]; European cultivated radish: ECR, which includes ESR, European oil radish [14], European RTR [3], BR and EPCR accessions. All radish: AR.

**Table 2 biology-10-01250-t002:** Population differentiation within the overall radish population and cultivated radish populations.

Indexes	OP	WCR	ACR	ECR
Fis	0.60	0.58	0.46	0.56
Fit	0.68	0.65	0.57	0.68
Fst	0.21	0.18	0.19	0.28
Nm	1.07	1.17	1.04	0.63

## Data Availability

Not applicable.

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
