# Peer review of "SSR-Sequencing Reveals the Inter- and Intraspecific Genetic Variation and Phylogenetic Relationships among an Extensive Collection of Radish (Raphanus) Germplasm Resources"

_biology, 2021, doi:10.3390/biology10121250_

Round 1
Reviewer 1 Report
In this manuscript, SSR sequencing method was used to detect the inter- and intraspecific genetic variation and phylogenetic relationships among 939 radish wild, semi-wild and cultivated accessions. Previous researchers have used classical molecular marker technology to reveal the diversity of radish germplasm resources, but this work is more accurate and comprehensive. The current study provides new perspectives into the origin, evolution and genetic diversity of Raphanus and facilitates the conservation and exploitation of radish germplasm resources.
There are several concerns about the research design and data, see below.
1, In Table 1, for Na index, the Average value is 5.95, however the Max value is 11.16, and the Min value is 1.92, this indicates that the Na index may not be appropriate for this study.
2, It is suggested to add an integration figure to show the phenotypes of different Radish.
Author Response
Dear reviewer:
Thank you very much for the positively opinions on my manuscript, we have corrected the manuscript based on your comments, Thankyou agagin.
Your sincerely
Xiaoman Li

Reviewer 2 Report
Dear authors,
I think you reported a well-crafted study. I have no major concerns with it. However, I have got a rather large number of minor concerns and suggestions to improve the quality of your manuscript (please see my comments in the attached pdf file). Please take care of meticulous responses to all these comments and suggestions. I assume that the final decision by editors on your submission will be based on the quality of your responses to the comments of reviewers, including those from me.
Best regards,

Author Response

(The authors gave the same response as above.)

Reviewer 3 Report
Review of Biology-1446288
SSR sequencing reveals the inter- and intraspecific genetic variation and phylogenetic relationships among an extensive collection of radish (Raphanus) germplasm resources
Xiaoman Li, Jinglei Wang, Yang Qiu, Haiping Wang, Peng Wang, Xiaohui Zhang, Caihua Li, Jiangping Song, Wenting Gui, Di Shen, Wenlong Yang, Bin Cai, Liwang Liu and Xixiang Li
The authors wished to study genetic variation and phylogenetic relationships in radishes. To do so they performed SSR sequencing on 939 wild, semi-wild and cultivated accessions. They found that European wild radishes had the highest genetic diversity and were separated from the domesticated varieties. They therefore concluded that Europe was the center of origin and that Europe, South Asia and East Asia appear to have been three independent centers of domestication. They also determined the population structures and prepared a phylogram for all 939 genotypes based on their 38 SSR genetic markers. In addition they determined the population differentiation and gene flow for the various groups of radishes.
Overall, the study seems to have been performed competently, the analyses seem appropriate and useful, and the results are interesting and worth sharing with the scientific community.
However, I have concerns about the amount of replication. It is unclear from the materials and methods, or from the figure captions and table legends, how many biological and how many technical replicates were performed on each accession. It is imperative to know how many biological replicates were sampled from each accession, and how many technical replicates were performed on each DNA sample. This must be corrected before this data can be published.
Otherwise, the study seems solid. The English is very good, with only a few minor problems as noted below.
Line 30: What is “multi-analysis?”
Lines 41-42 are grammatically incorrect and hard to understand.
Lines 47-48 are grammatically incorrect and hard to understand.
Line 88: “Japan” should be “Japanese.”
Lines 111-112: please explain better! SSRs are 2-5 bp sequences that are repeated nose-to-tail, and the number of copies at each locus is what varies.
Line 176: what is the concentration of the primers and of the genomic DNA?
Line 182: What are “polymorphic primers?”
Line 282: Table 1: I concede that the table is already crowded, but it would be helpful to indicate the variability for each of these numbers, e.g. state the number ±s.dev or s.e. in order to indicate how reliable these numbers are.
Lines 334-336 are grammatically incorrect and hard to understand.
Lines 345-350 are grammatically incorrect and hard to understand.
Author Response

(The authors gave the same response as above.)
